# EGCG Disrupts the LIN28B/Let-7 Interaction and Reduces Neuroblastoma Aggressiveness

**DOI:** 10.3390/ijms25094795

**Published:** 2024-04-27

**Authors:** Simona Cocchi, Valentina Greco, Viktoryia Sidarovich, Jacopo Vigna, Francesca Broso, Diana Corallo, Jacopo Zasso, Angela Re, Emanuele Filiberto Rosatti, Sara Longhi, Andrea Defant, Federico Ladu, Vanna Sanna, Valentina Adami, Vito G. D’Agostino, Mattia Sturlese, Mario Sechi, Sanja Aveic, Ines Mancini, Denise Sighel, Alessandro Quattrone

**Affiliations:** 1Department of Cellular, Computational and Integrative Biology (CIBIO), University of Trento, 38123 Trento, Italy; simonacocchi90@gmail.com (S.C.); valentina.greco@unitn.it (V.G.); viktoryia.sidarovich@unitn.it (V.S.); vito.dagostino@unitn.it (V.G.D.); 2Department of Physics, University of Trento, 38123 Trento, Italy; andrea.defant@ex-staff.unitn.it (A.D.);; 3Istituto di Ricerca Pediatrica Fondazione Città della Speranza, 35127 Padova, Italy; 4Department of Medicine, Surgery and Pharmacy, University of Sassari, 07100 Sassari, Italy; fladu@uniss.it (F.L.); mario.sechi@uniss.it (M.S.); 5Nanomater S.r.l., 07041 Alghero, Italy; 6Molecular Modeling Section, Department of Pharmaceutical and Pharmacological Sciences, University of Padua, 35127 Padova, Italy; mattia.sturlese@unipd.it

**Keywords:** (−)-epigallocatechin 3-gallate, EGCG, LIN28B/let-7 interaction inhibitors, neuroblastoma, AlphaScreen, PLC/PLGA-PEG nanoparticles, differentiation therapy, target therapy

## Abstract

Neuroblastoma (NB) is the most commonly diagnosed extracranial solid tumor in children, accounting for 15% of all childhood cancer deaths. Although the 5-year survival rate of patients with a high-risk disease has increased in recent decades, NB remains a challenge in pediatric oncology, and the identification of novel potential therapeutic targets and agents is an urgent clinical need. The RNA-binding protein LIN28B has been identified as an oncogene in NB and is associated with a poor prognosis. Given that LIN28B acts by negatively regulating the biogenesis of the tumor suppressor let-7 miRNAs, we reasoned that selective interference with the LIN28B/let-7 miRNA interaction would increase let-7 miRNA levels, ultimately leading to reduced NB aggressiveness. Here, we selected (−)-epigallocatechin 3-gallate (EGCG) out of 4959 molecules screened as the molecule with the best inhibitory activity on LIN28B/let-7 miRNA interaction and showed that treatment with PLC/PLGA-PEG nanoparticles containing EGCG (EGCG-NPs) led to an increase in mature let-7 miRNAs and a consequent inhibition of NB cell growth. In addition, EGCG-NP pretreatment reduced the tumorigenic potential of NB cells in vivo. These experiments suggest that the LIN28B/let-7 miRNA axis is a good therapeutic target in NB and that EGCG, which can interfere with this interaction, deserves further preclinical evaluation.

## 1. Introduction

Neuroblastoma (NB) is the most frequently diagnosed extracranial solid tumor in children. The majority of cases occur in patients under the age of 5, with an average age of diagnosis of 2 years [1]. Despite being considered a relatively rare disease affecting 1 in 7000 live births, NB accounts for 15% of all pediatric cancer-related deaths [1,2,3].

NBs are highly heterogeneous tumors in terms of clinical presentation and outcome. Indeed, they include cases that spontaneously regress even if metastatic, low-to-intermediate-risk cases with tumors that can be surgically resected and treated by chemotherapy as well as high-risk cases, which are often metastatic and treatment-refractory [1,4,5]. Although the 5-year survival rate of patients with the high-risk disease has increased from less than 20% to over 50% in the past few decades [1], NB remains a challenge in pediatric oncology, and the identification of novel potential therapeutic targets and agents is an urgent clinical need.

NBs arise from the developing peripheral sympathetic nervous system in the adrenal medulla or along the sympathetic chain [6,7]. Specifically, NBs originate from the transient population of neural crest cells that undergo defective sympathetic neuronal differentiation [6,7]. Being a developmental tumor, NBs have a few associated genetic mutations. The most common genetic alteration found in NBs is the focal amplification of the *MYCN* gene, which is present in 25% of cases [8,9], while other frequently recurring mutations are located in the *ALK* [10,11] and *ATRX* [12] genes. Additionally, overexpression of the *TERT* [13,14] and *LIN28B* [15,16,17] genes has been associated with NB onset.

LIN28 is an evolutionarily conserved RNA-binding protein that was first characterized in *Caenorhabditis elegans* due to its crucial role in development and the regulation of developmental timing [18]. In mammals, the two LIN28 paralogs, LIN28A and LIN28B, which act as gatekeepers regulating the transition between pluripotency and committed cell lineages, are highly expressed in the early developmental stages, decrease upon differentiation, and are typically absent in most differentiated cells in adults [19]. Reactivation of either LIN28A or LIN28B is common in many human cancers, where their expression is usually mutually exclusive [20,21]. The paralog LIN28B is recognized as an oncogene in NB and plays an important role in NB tumorigenesis [15,16,17]. Indeed, forced expression of LIN28B in nude mice is sufficient to induce NB formation [16]. Furthermore, a high expression of LIN28B in NB is associated with a poor prognosis, an aggressive disease phenotype, and the promotion of tumor cell migration and survival [15,16,17].

Even if the mechanisms by which LIN28B drives tumor development and progression are not completely understood, it is well-known that LIN28B acts as a negative regulator of the biogenesis of the tumor suppressors let-7 miRNAs [22,23]. Specifically, LIN28B selectively blocks the processing of let-7 miRNA precursor molecules into mature miRNAs, resulting in lower amounts of mature let-7 miRNAs [22,23], which in turn cannot exert their tumor suppressor activity by directly repressing several well-known oncogene targets, including RAS, MYC, HMGA2, and BLIMP1 [20,21].

Given the critical role of the LIN28B/let-7 miRNA pathway in NB, we reasoned that selective interference with the LIN28B/let-7 miRNA circuit would result in increased levels of let-7 miRNAs and, as a consequence, decreased cell proliferation and induction of cell differentiation, ultimately reducing NB aggressiveness.

In the work presented here, we exploited two orthogonal biochemical techniques to screen and validate 4959 molecules and selected (−)-epigallocatechin 3-gallate (EGCG) as the molecule with the best inhibitory activity on the LIN28B/let-7 miRNA interaction. Since EGCG is unstable under cell culture conditions, we encapsulated it in PLC/PLGA-PEG polymeric nanoparticles (EGCG-NPs), which are a well-studied nanocarrier system due to their high biocompatibility, high drug encapsulation rate, and suitability for targeted therapy. We showed that EGCG-NP treatment led to a strong increase in mature let-7 miRNAs and a consequent inhibition of growth and promotion of differentiation in NB cells. Finally, we also showed that EGCG-NP pretreatment reduced the tumorigenic capacity of NB cells using a zebrafish xenograft model. Taken together, these experiments suggest that the LIN28B/let-7 miRNA axis is a valuable therapeutic target in NB and that EGCG, which can interfere with this interaction, deserves further preclinical evaluation.

## 2. Results

### 2.1. LIN28B Downregulation Increases Let-7 miRNA Levels and Reduces Aggressiveness in NB Cells

To characterize the effects of modulating the LIN28B/let-7 interaction on the NB cell phenotype, we analyzed four NB cell lines for LIN28B and MYCN expression levels and downregulated LIN28B expression in two of them, namely CHP134 and NB69, which express different levels of LIN28B and have different *MYCN* amplification status, being *MYCN*-amplified and *MYCN*-non-amplified, respectively [24] (Figure 1A). LIN28B downregulation (Figure 1B–D) led to a statistically significant increase in almost all let-7 miRNA family members tested (let-7d, let-7f, let-7g, let-7i) in both cell lines (Figure 1E), as previously observed by other groups [16,22,23]. Next, we investigated whether the observed downregulation of LIN28B and the increase in let-7 miRNA levels would lead to a reduction in the aggressiveness in NB cells. To this end, we examined the mRNA and protein levels of several tumor and differentiation markers. Stable LIN28B downregulation resulted in a significant decrease in the tumor markers SOX2, NESTIN, and SOX9 and an increase in the neural differentiation marker GAP43 (Figure 1F–H), consistent with the critical role of LIN28B in maintaining stemness.

### 2.2. A High-Throughput Screen Identifies Candidate Molecules Capable of Interfering with the LIN28B/Let-7 Interaction

To identify small molecules capable of targeting the LIN28B/let-7 interaction, we screened two commercial libraries containing a total of 4959 molecules, including FDA-approved drugs, natural products, and drug-like compounds, using the Amplified Luminescent Proximity Homogeneous Assay (AlphaScreen) technique. For the assay, we used a biotinylated precursor let-7g (pre-let-7g) miRNA, which binds to streptavidin-coated beads, and a c-MYC-tagged recombinant LIN28B protein (rLIN28B), which binds to anti-c-myc acceptor beads. Upon excitation at 680 nm, an emission of light at 570 nm is observed when the donor and acceptor beads are in close proximity due to the interaction between the miRNA and the protein. Conversely, disruption of LIN28B/let-7 binding results in a reduction or absence of the signal (schematic representation in Figure 2A, see Appendix A for assay setup). Based on the results of the primary screen (Figure 2B, Z factor of 0.65 and a signal-to-background ratio of 12.49) and of a subsequent confirmatory screen, we selected 29 compounds (0.58% of the original compounds tested, Appendix A), which we orthogonally validated using the RNA electrophoretic mobility shift assay (REMSA, schematic representation in Figure 2C). Specifically, we incubated a cyanine-3-labeled pre-let-7g miRNA, the rLIN28B protein, and the molecule of interest for 1 h and then subjected them to electrophoresis on a gel. As a positive control, we used a free cyanine-3-labeled pre-let-7g miRNA without the addition of the purified rLIN28B protein. Figure 2D shows representative images of the REMSA performed to validate the molecules selected after the AlphaScreen. For molecules capable of interfering with the binding between the protein and the RNA probe, the protein/RNA complex does not form, and the free miRNA probe is detected in the lower part of the gel.

Based on the results of the REMSA, we finally selected four hits as candidate inhibitors: (−)-epigallocatechin 3-gallate (EGCG), theaflavin monogallates (TFMG), gallic acid (GA), and aurintricarboxylic acid (ATA), whose molecular structures are depicted in Figure 2E. Of note, TFMG was present in the screened library as mixed isomers (theaflavin 3-gallate and theaflavin 3′-gallate) from black tea. Interestingly, EGCG and TFMG are characterized by a high degree of structural similarity (highlighted in pink in Figure 2E), while the third hit (GA) is a small molecule whose structure is completely encompassed by those of EGCG and TFMG. The fourth hit, although already reported as a LIN28B inhibitor as a result of a drug screen performed by another group using a different approach [25], was not considered for further evaluation due to its intrinsic properties. Indeed, ATA has been reported to readily polymerize in an aqueous solution, forming a stable free radical that inhibits protein/nucleic acid interactions, and is known to be a promiscuous pan-selective inhibitor of DNA and RNA processing enzymes, presumably due to its DNA-mimetic properties [26].

### 2.3. EGCG Interferes with the LIN28B/let-7 Interaction by Binding to LIN28B

We then measured the extent of the interference of EGCG, TFMG, and GA in dose-dependent titration experiments using the Alpha assay. Due to its structural similarity to EGCG and TFMG, we also included (−)-epigallocatechin 3,5-digallate (EGCDG), which emerged from the primary screen but was not subsequently confirmed (molecular structure shown in Figure 3A; the structural similarity with EGCG is highlighted in pink). All the compounds tested, except for EGCDG, effectively inhibited the association between LIN28B and the pre-let-7g miRNA, with an inhibition constant (K_i_) in the low nanomolar range and with EGCG and GA slightly more potent than TFMG (Figure 3B). Next, to test for possible broad in vitro effects of the EGCG and TFMG, we challenged these compounds with another well-characterized protein/RNA interaction, which occurs between the RNA-binding protein HuR and the TNFα AU-rich element, and for which other small molecule inhibitors have been described [27]. As expected, EGCG and TFMG did not affect HuR/RNA binding (Figure 3C).

To further investigate the binding of the hits to LIN28B, we then performed a structure-based molecular modeling study. Specifically, we used three experimentally solved structures to generate a 3D model of LIN28B by homology modeling. This model includes the protein’s cold shock domain and the zinc knuckle domain (Figure 3D). Docking calculations indicated that both EGCG and TFMG have good steric complementarity with the pre-let-7 miRNA binding site of LIN28B (Figure 3E,F) with ChemPLP scores of −79.6 and −73.1, respectively. Both hits place their gallate moiety into the pocket formed by Tyr130, Lys149, Lys150, Cys151, His152, Met160, and Val161, establishing hydrophobic contacts, electrostatic interactions, and hydrogen bonds with this pocket. The docking prediction also suggested that the common gallate moiety forms a hydrogen bond with the backbone of Lys150. Moreover, the catechin scaffold is stabilized by π–π stacking with Tyr130 and hydrophobic contacts with the methylene moieties of Lys121 and Lys123. EGCDG also showed a similar interaction pattern to EGCG and TFMG but with the lowest ChemPLP score of −71.6, suggesting a lower complementarity to the binding site (Figure 3G). Although GA assumed the same confirmation observed for the gallate moiety in EGCG, TFMG, and EGCDG, its reduced chemical complexity resulted in different conformations with similar scores, making it more difficult to rationalize (Figure 3H).

Based on these results, we selected EGCG for further studies as it guarantees a good binding mode and a better in silico binding efficiency than TFMG and EGCDG.

### 2.4. EGCG Encapsulated in Nanoparticles Affects NB Cell Viability

Since EGCG has been reported to have low stability and to oxidize under cell culture conditions [28,29], we evaluated the stability of EGCG under our cell culture conditions using the high-performance liquid chromatography (HPLC) analysis. Specifically, after constructing a calibration curve (Appendix A), we analyzed a 50 µM solution of EGCG dissolved in cell culture media at different time points. The HPLC analysis clearly showed that EGCG is unstable under biological test conditions, being reduced by more than 50% after 15 min and completely degraded after 45 min (Figure 4A and Appendix A). Therefore, to enhance the stability and deliverability of EGCG into cells, we encapsulated the molecule inside polymeric nanoparticles, which are a well-studied biodegradable and biocompatible drug nanocarrier system [30,31]. Specifically, we used a blend of poly(epsilon)-caprolactone and poly-lactide-co-glycolide-polyethylene glycol (PCL/PLGA-PEG)-based nanoparticles (NPs) (Figure 4B), which exhibited the highest performance in terms of EGCG loading content, encapsulation efficiency, and production yields [32].

To assess whether NPs penetrate and effectively accumulate in NB cells, we first treated NB69 cells with different amounts of NPs containing the fluorophore coumarin 6 (Cou6-NPs) and analyzed them using the Operetta High Content Imaging System. As expected, the fluorophore signal contained in the NPs was localized inside the cells (Appendix A), and the fluorescence intensity increased proportionally with the concentration of NPs used (Figure 4C). To better visualize the Cou6-NPs inside the cells, we then treated CHP134 cells with 0.003 µg/µL of Cou6-NPs and imaged them using a confocal microscope. Again, this analysis showed that all the cells analyzed had internalized the NP content (Figure 4D), with no Cou6 fluorescence signal in the nuclei, indicating that the NP content is released into the cytoplasm.

We then evaluated the effect of EGCG-containing NPs (EGCG-NPs) on three NB cell lines. Specifically, we used the CHP134 and NB69 cell lines that we used for LIN28B downregulation and the KELLY cell line, which harbors extremely high amplification levels of the MYCN locus [24] and expresses high levels of LIN28B (Figure 1A). By treating the cells with different concentrations of NPs for 48 h, we assessed the effects of EGCG-NPs on cell viability. To ensure the observed effect was not due to the polymeric carrier alone, we added controls with the same increasing concentrations of EGCG-free NPs (empty-NPs). The empty-NPs showed no significant toxicity, even at the highest concentrations, while the three cell lines showed varying degrees of sensitivity to EGCG-NPs (Appendix A). We then calculated the viability half-maximal inhibitory concentration (IC_50_) values, which is defined as the compound concentration that causes 50% inhibition of cell viability. Nanoencapsulation resulted in a significant decrease in EGCG IC_50_ values in all cell lines tested compared to those treated with non-encapsulated EGCG (Figure 4E), suggesting that the polymeric matrix stabilizes the EGCG in the solution and leads to higher concentrations of EGCG inside the cells.

### 2.5. EGCG-NP Treatment Decreases In Vitro Proliferation and Stemness and Reduces Tumorigenic Potential of NB Cells in a Zebrafish Model

We then proceeded to assess the molecular effects of EGCG-NP treatment in NB cells using doses of EGCG-NPs below or around the previously determined IC_50_ values (Figure 4E). Specifically, we first assessed the effects on the levels of three let-7 miRNA family members (let-7d, let-7f, let-7g) by qPCR. Treatment with EGCG-NPs significantly increased let-7 miRNA levels in all cell lines tested, with a dose-dependent effect particularly evident in KELLY cells (Figure 5A).

At the same time, EGCG-NP treatment also resulted in a dose-dependent reduction in NB cell proliferation (Figure 5B), while empty-NPs only slightly affected cell growth (Appendix A). Furthermore, EGCG-NP treatment in CHP134 cells, which have been reported to be prone to differentiation upon specific stimuli, such as 13-cis-retinoic acid treatment [33], led to a significant decrease in the mRNA levels of the tumor markers *MYCN*, *SOX2*, and *SOX9* and a significant increase in the differentiation markers *TUBB3*, *GAP43,* and *TH* (Figure 5C).

Since the increase in let-7 miRNA levels, decrease in proliferative capacity, and promoted differentiation observed after EGCG-NP treatment are expected to affect the aggressiveness of NB cells, we next investigated whether EGCG-NP pretreatment would affect the tumorigenic potential of NB cells in vivo. To this end, we injected fluorescently labeled NB cells, pretreated for 48 h with either empty-NPs or EGCG-NPs, into the duct of Cuvier of 48-h-old Tg(fli1:EGFP) zebrafish embryos, which express the green fluorescent protein throughout the vascular endothelium, allowing for visualization of blood vessels [34]. Since CHP134 cells do not possess a strong engrafting ability, and no data are available for KELLY and NB69 cells in the literature, we used the SK-N-BE(2) cell line, whose engraftment capacity has already been reported [35]. As for the NB69, KELLY, and CHP134 cell lines (Appendix A), SK-N-BE(2) also showed a dose-dependent decrease in viability (Appendix A) and an increase in the let-7 miRNA levels (Appendix A) upon in vitro EGCG-NP treatment. Immediately after in vivo injection, pre-treated SK-N-BE(2) cells rapidly spread throughout the embryonic vasculature, mimicking the metastatic spread of the disease (Figure 5D, top panel). At 24 h post-injection, the cells were found arrested along the endothelium of the trunk and the tail regions of the embryos (Figure 5D, bottom panels). Thus, we evaluated the maintenance of the fluorescence intensity in the caudal region of each injected embryo deriving from NB cells pretreated with either EGCG-NPs or empty-NPs. In both conditions, we observed a lower fluorescence intensity signal compared to time 0 (Figure 5D,E), but with a much greater decrease for EGCG-NP-pretreated cells. Specifically, after 24 h, the fluorescence intensity signal of empty-NP-pretreated cells decreased by approximately 50%, while the signal of EGCG-NP-pretreated cells decreased to 3.4%, suggesting that EGCG-NP pretreatment strongly reduces the tumorigenic potential of NB cells in vivo.

Taken together, these results indicate that EGCG-NP treatment effectively reduced NB cell aggressiveness and their tumorigenic potential in vivo.

## 3. Discussion

Reactivated LIN28A or LIN28B expression almost invariably correlates with poor prognosis in many tumors, including acute myeloid leukemia [36], brain cancers [37,38], and NB [15,16,17]. Disruption of the LIN28B/let-7 miRNA interaction restores the tumor suppressor let-7 miRNA levels, providing a potential new therapeutic target in oncology [20,21]. Indeed, identifying small molecules capable of interfering with the LIN28B/let-7 interaction holds great promise for the development of new anticancer treatments. In this work, using two orthogonal biochemical techniques, we screened 4959 molecules and selected (−)-epigallocatechin 3-gallate (EGCG) as the molecule with the best inhibitory activity on the LIN28B/let-7 miRNA interaction. Interestingly, TFMG, which shares the same epigallocatechin scaffold as EGCG, emerged as a second hit from our screening and subsequent validation. Our results complement those of several screens performed in recent years, which have led to the identification of some other molecules capable of disrupting the LIN28B/let-7 interaction ([25,39,40,41,42,43], reviewed in [44]). Docking calculations indicate that EGCG binds to LIN28B in the pre-let-7 miRNA binding pocket in the cold shock domain of the protein, forming hydrophobic contacts, electrostatic interactions, and hydrogen bonds with this pocket. Further interactions are also formed through the gallic acid moiety and the catechin backbone. 

EGCG is an abundant polyphenolic component of green tea extract and has been reported to possess various biological functions, including antioxidant, anti-inflammatory, and anticancer properties [45,46,47]. The anticancer properties of EGCG have been linked to several important cellular signaling pathways, including those mediated by EGFR, JAK-STAT, MAPKs, NF-κB, and PI3K-AKT-mTOR [45,47]. Moreover, EGCG has been shown to bind to and inhibit the human peptidyl prolyl cis/trans isomerase (Pin1), which plays a critical role in oncogenic signaling [48,49]. Interestingly, EGCG has also been reported to upregulate let-7 miRNAs in human lung cancer and melanoma cells [50,51], consistent with possible LIN28B inhibition also in these cancer models.

Despite a wide range of reported potential therapeutic and promising results in preclinical studies, EGCG is known to have low stability under cell culture conditions [28]. Furthermore, the applicability of EGCG in humans has been hampered by its low bioavailability, poor membrane permeability, rapid metabolic clearance, and lack of stability [52]. Based on this evidence, we decided to incapsulate EGCG into suitable nanocarriers. 

In recent years, the application of nanoparticles for drug encapsulation in cancer has gained increasing interest due to the potential to improve their delivery and pharmacokinetic properties while reducing the overall toxicity of treatments [53,54]. To date, most studies on the application of the nanomedicine strategy to NB therapeutics have been conducted mainly in the preclinical setting using cellular and animal experiments, which together have provided some positive evidence [55,56]. Despite these encouraging results, the therapeutic potential of nanomedicine in NB has not yet been systematically explored, and only albumin-bound paclitaxel nanoparticles (i.e., Abraxane) have reached phase I/II clinical trials for refractory NB and other pediatric solid tumors (NCT01962103) [57,58].

In this scenario, among the different approaches pursued in the field of nanoformulation, we chose to use polymeric nanoparticles as a model of nanosystems for ECGC delivery. Polymeric nanoparticles are a well-studied nanocarrier system due to their high biocompatibility, high versatility, and high encapsulation rate [30,31]. We have previously shown that encapsulation of EGCG into polymeric blended nanosystems, targeted with small molecules capable of binding to the prostate-specific membrane antigen, enhances the antiproliferative activity of EGCG in prostate cancer both in vitro and in vivo [32].

Here, we used non-targeted NPs to deliver EGCG into NB cells. EGCG encapsulation resulted in a significant decrease in IC_50_ values compared to the free drug in all cell lines analyzed, suggesting that the PLC/PLGA-PEG nano-construct protects and stabilizes EGCG, leading to a significant increase in molecule accumulation within the cells. In addition, EGCG-NP treatment resulted in a significant increase in all let-7 miRNA family members analyzed, demonstrating that EGCG effectively interferes with the LIN28B/let-7 miRNA interaction. Of note, LIN28B has been shown to have pro-tumorigenic activity independent of its interaction with let-7 miRNAs by binding to specific mRNAs and acting as a post-transcriptional regulator [21]. Further experiments to investigate whether EGCG can also affect LIN28B pro-tumorigenic activity in a let-7-independent manner are definitely needed and may strengthen the relevance of EGCG in NB and potentially other tumor types.

The increase in mature let-7 miRNA levels upon EGCG-NP treatment was also accompanied by dose-dependent inhibition of cell growth, decreased tumor marker expression, and increased differentiation marker expression. Given the role of let-7 miRNAs, which have been described as fundamental tumor suppressors and essential regulators of terminal differentiation [59], and considering the effects we observed with EGCG treatment, EGCG may promote NB cell differentiation. Finally, EGCG-NP treatment also significantly reduced the tumorigenic potential of NB cells in a zebrafish xenograft model.

The poor pharmacokinetic profile of EGCG, which requires the use of a nanocarrier-based formulation, represents the main limitation of the present study and potentially hinders a streamlined clinical development. Future studies aimed at elucidating the structure of EGCG bound to LIN28B could guide the design of new molecules with greater potency and better pharmacokinetic properties, with the ultimate aim of selecting a molecule more suitable for further preclinical and clinical development. In this context, EGCG should be considered as a chemical probe to prove that inhibition of the LIN28B/let-7 axis is a novel and promising therapeutic option for NB, especially with regard to the development of new agents for differentiation therapies.

Therapies capable of inducing cancer cell differentiation have long been considered an alternative to cytotoxic therapy to suppress tumorigenesis, but differentiation therapy is still a largely unexplored field. The most successful example of differentiation therapy in the clinic is the combination of the differentiation-inducing agents all-trans-retinoic acid and arsenic trioxide, which has led to clinical complete remission rates of over 90% in patients with acute promyelocytic leukemia [60]. In NB, 13-cis-retinoic acid, a pro-differentiating agent, is currently used in clinical practice as part of the treatment of patients with high-risk NB in the post-consolidation phase of the therapeutic schedule [61,62,63]. Indeed, the rate of tumor relapse is directly dependent on the efficacy of post-consolidation. Unfortunately, many patients are refractory to retinoic acid-induced differentiation and further research efforts, including synergistic combination therapy, are needed [64,65]. In this context, EGCG and, more generally, inhibitors of the LIN28B/let-7 miRNA circuit may represent good candidates to be used as experimental drugs in the post consolidation phase for high-risk patients, both alone and as part of synergistic pro-differentiating multi-drug treatments. Additional in vitro and in vivo studies are required to further explore the potential clinical use of LIN28B inhibitors in NB.

## 4. Materials and Methods

### 4.1. Cell Cultures

Human NB cell lines were purchased from the European Collection of Authenticated Cells (ECACC, Porton Down, Salisbury, UK). NB69 (cat. 99072802, ECACC), KELLY (cat. 92110411, ECACC), and CHP134 (cat. 06122002, ECACC) cells were cultured in RPMI-1640 (cat. 11875093, Thermo Fisher Scientific, Waltham, MA, USA) supplemented with 2 mM of glutamine (cat. A2916801, Thermo Fisher Scientific), 10% fetal bovine serum (FBS) (cat. 10270106, Thermo Fisher Scientific), and 1% penicillin-streptomycin (10,000 U/mL penicillin, 10000 μg/mL streptomycin, cat. 15140122, GIBCO, Thermo Fisher Scientific) at 37 °C and 5% CO_2_.

Human NB SK-N-BE(2) cells (cat. 95011815, ECACC) were cultured in a 1:1 mixture of EMEM/F-12 (cat. 670086/11765054, Thermo Fisher Scientific) supplemented with 2 mM of glutamine, 10% FBS, 1% penicillin-streptomycin, and 1% non-essential amino acids (cat. 11140050, Thermo Fisher Scientific) at 37 °C and 5% CO_2_.

Human embryonic kidney HEK293T cells were obtained from the Interlab Cell Line Collection (ICLC) (cat. HTL04001, IRCCS Ospedale Policlinico San Martino, Genova, Italy) and were cultured in DMEM (cat. 11960044, Thermo Fisher Scientific) supplemented with 2 mM of glutamine, 10% FBS, and 1% penicillin-streptomycin at 37 °C and 5% CO_2_.

### 4.2. LIN28B Downregulated Cell Line Generation

Lentiviral particles were produced in HEK293T cells by transfecting 10 µg of LIN28B-shRNA or scramble-shRNA (referred to as shCTRL in the main text) inserted into MISSION^®^ pLKO.1-puro Empty Vector Control Plasmid DNA (cat. SHC001, Sigma-Aldrich, St. Louis, MO, USA) with the packaging vectors psPAX2 (5 µg, cat. 12260, Addgene, Watertown, MA, USA) and pMD2.G (2.5 µg, cat. 12259, Addgene) in serum-free Opti-MEM (cat. 31985070, GIBCO, Thermo Fisher Scientific). Lipofectamine 2000 (cat. 11668500, Thermo Fisher Scientific) was used as a transfecting agent in a 1:1 ratio with the plasmid mixture. Supernatants were harvested 48 h later, filtered through a 0.45 µm filter, and the produced viral particles were concentrated by ultracentrifugation. Viral particles were aliquoted and stored frozen at −80 °C.

LIN28B-stably-downregulated CHP134 and NB69 cells were generated by transducing CHP134 and NB69 cells with the viral particles containing LIN28B shRNA or scramble shRNA for 8–10 h. After 24 h, the transduced cells were selected by supplementing media with 3–5 µg/mL puromycin (cat. ant-pr-5b, InvivoGen, San Diego, CA, USA) for at least 72 h.

### 4.3. RNA Extraction, Reverse Transcription, and qPCR

Total RNA was extracted using the Trizol^TM^ Reagent (cat. 15596026, Thermo Fisher Scientific) according to the manufacturer’s instructions. Reverse transcription was performed on 1 µg of RNA with the RevertAid RT Reverse Transcription Kit (cat. K1691, Thermo Fisher Scientific) on the C1000 Thermal Cycler (Bio-Rad, Hercules, CA, USA). The cDNA was diluted to 5 ng/µL, and qPCR was performed using the KAPA SYBR FAST qPCR Master Mix (2X) (cat. SFUKB, Kapa Biosystems, Wilmington, MA, USA) according to the manufacturer’s indications on the CFX96 Real-Time System (Bio-Rad). All assays were performed in triplicate in 3–4 independent experiments. Data were analyzed using the CFX Manager software 3.1 (Bio-Rad) and quantified using the ΔΔCt method. *HPRT1* or *SDHA* was used as a reference gene, and shCTRL cells, non-treated cells, or empty-NP cells were used as internal calibrators (as specified in the figure legend). Primer sequences can be found in Appendix A.

### 4.4. Immunoblotting

Total cell lysates were prepared from cells. Briefly, cells were washed with PBS and resuspended in a RIPA lysis buffer (cat. 89901, Thermo Fisher Scientific) supplemented with protease inhibitors. Protein concentrations were quantified with the Pierce^TM^ BCA Protein Assay Kit (cat. A55864, Thermo Fisher Scientific). Equal amounts of protein (25 µg) were separated on SDS-PAGE and transferred to a nitrocellulose membrane. Membranes were probed with anti-LIN28B (diluted 1:1000, cat. 4196, Cell Signaling, Danvers, MA, USA), anti-MYCN (diluted 1:1000, cat. 9405, Cell Signaling), anti-NESTIN (diluted 1:1000, cat. sc-23927, Santa Cruz Biotechnology, Dallas, TX, USA), anti-SOX9 (diluted 1:500, cat. 702016, Thermo Fisher Scientific), anti-GAP43 (diluted 1:500, cat. AB5220, Merck Millipore, Darmstadt, Germany), anti-GAPDH (diluted 1:1000, cat. sc-32233, Santa Cruz Biotechnology), anti-β-TUBULIN (diluted 1:3000, cat. sc-53140, Santa Cruz Biotechnology), and secondary HRP-conjugated antibodies (diluted 1:3000, cat. 62-6520 and 31460, Invitrogen). Primary antibodies were probed overnight at 4 °C, while secondary antibodies were probed for 1 h at room temperature. Detection was performed using Amersham ECL Prime or the Select Western Blotting Detection Reagent (cat. RPN2232 or cat. RPN2235, GE Healthcare Life Sciences, Chicago, IL, USA) and the ChemiDoc Imaging System (Bio-Rad). Data were analyzed using Image Lab™ Software, Version 3.0.

### 4.5. Let-7 miRNA Quantification

Total RNA was extracted using the Trizol^TM^ Reagent (cat. 15596026, Thermo Fisher Scientific) according to the manufacturer’s instructions. The miRCURY LNA RT Kit (cat. 339340, Qiagen, Hilden, Germany) was used to perform the reverse transcription step following the manufacturer’s instructions, while the miRCURY LNA SYBR Green PCR Kit (cat. 339345, Qiagen) was used to perform the qPCR. The qPCR was run using the CFX96 Real-Time System (Bio-Rad). The data were analyzed with CFX Manager software 3.1 (Bio-Rad) and normalized on U6 content. The following miRCURY LNA miRNA PCR Primer mixes (cat. 339306, Qiagen) were used: U6 snRNA (YP00203907); hsa-let-7d-5p (YP00204124); hsa-let-7f-5p (YP00204359); hsa-let-7g-5p (YP00204565); hsa-let-7i-5p (YP00204394).

### 4.6. rLIN28B Protein Expression and Purification

The full-length human LIN28B cDNA sequence (NM_001004317.3) was amplified from retro-transcribed RNA of HEK293 cells and inserted into the pCMV6-AC-Myc-His mammalian expression vector (cat. PS100006, Origene Technologies, Rockville, MD, USA) by using the forward (5′ AGTCGCGATCGCATGGCCGAAGGCGGGGC 3′) and reverse (5′ ACGTACGCGTTGTCTTTTTCCTTTTTTGAACTGAAGGCC 3′) primers containing the SgfI and the MluI restriction sites, respectively. The frame and sequence of the full-length open reading frame in the newly cloned vector, hereafter named pCMV6-LIN28B-Myc-His, were confirmed by sequencing. Recombinant human LIN28B-Myc-His protein (rLIN28B) was produced by transient transfection of HEK293T cells with the pCMV6-LIN28B-Myc-His vector using polyethyleneimine (PEI, cat. 408727, Sigma-Aldrich, vector/PEI ratio = 1:3). Briefly, cells were harvested 24 h post-transfection in the EQ buffer (see Appendix A for buffer composition) and sonicated (amplitude of 45, 7 cycles of 10 s, 10 s pause between each cycle, power at approximately 250 W) at 4 °C. The rLin28B protein was purified using Ni-NTA agarose beads (cat. 30210, Qiagen) and eluted with an imidazole gradient ranging from 10 to 400 mM. The protein was dialyzed using D-Tube™ Dialyzers midi (cat. 71506, Merck Millipore) for 2 h at 4 °C, aliquoted, and stored at −80 °C in buffer S.

The rLin28B protein was analyzed by Coomassie staining on 15% SDS-PAGE. The relative protein concentration was determined using bovine serum albumin standards and densitometric quantification of the corresponding bands on acrylamide gels. Western blot analysis was performed using a monoclonal anti-Myc antibody (diluted 1:1000, overnight incubation at 4 °C, cat. TA150014, Origene).

### 4.7. Amplified Luminescent Proximity Homogeneous Assay (AlphaScreen)

The AlphaScreen assay was performed following the manufacturer’s instructions (PerkinElmer, Waltham, MA, USA). Specifically, we used the rLIN28B protein and a 5′-biotinylated single-stranded RNA corresponding to the precursor of the let-7g miRNA (Bi-pre-let-7g miRNA, 5′-Bi-GCUAUGAUACCACCCGGUACAGGAGC 3′), whose interaction with LIN28B is mediated by the specific GGAG motif present on its terminal loop [66]. As a positive control, instead of the Bi-pre-let-7g miRNA, we used a Bi-pre-let-7g-mut miRNA, which is unable to bind to the LIN28B protein due to the absence of the adenine in the conserved motif essential for the interaction, simulating the absence of interaction between LIN28B and the pre-let-7g miRNA (5′-Bi-GGCAUGAUACCACCCGGUACGGGC 3′). RNA probes were purchased from Eurofins MWG Operon. The assay was performed in a dialysis buffer in 384-well white OptiPlates (cat. 6007299, PerkinElmer) in a final volume of 25 µL using the AlphaScreen c-Myc detection kit (cat. 6760611M, PerkinElmer). The optimal concentrations for the two interacting partners were determined by titration as 10 nM and 100 nM for rLIN28B and for Bi-pre-let-7g miRNA, respectively. Anti-c-Myc-acceptor beads and streptavidin-donor beads (cat. 6760002S, PerkinElmer) (10 µg/mL final concentration) were added to each well containing a compound to be tested, the rLIN28B protein and the Bi-pre-let-7g miRNA, and the reaction was incubated at room temperature for 90 min. Two commercial libraries (MicroSource Spectrum Collection, MicroSource Discovery Systems, Gaylordsville, USA and NDL-3000, TimTec, Tampa, USA) containing a total of 4959 molecules were screened in the primary screening. Molecules were tested at the final concentration of 75 nM in monoplicates. All the dispensation steps were performed using Tecan EVO 200 (Tecan, Männedorf, Switzerland). Fluorescence signals were detected on the Enspire plate reader instrument (cat. 2300-001A, PerkinElmer), and the specific interaction signal was quantified by subtracting the background signal, calculated in the absence of the protein and/or the probe. Compounds that differed by 2 times the standard deviation from the mean of the negative controls were selected as hits. See Appendix A for the assay setup.

### 4.8. RNA-Electrophoresis Mobility Shift Assay (REMSA)

Six nM rLIN28B proteins and 6 nM of a Cy3-labeled pre-let-7g RNA probe (5′-Cy3-GCUAUGAUACCACCCGGUACAGGAGC 3′, Eurofins MWG Operon, Ebersberg, Germany) were incubated with 0.5 µM of the selected hits, for a final volume of 20 µL (20 mM HEPES pH 7.5, 50 mM KCl, 0.5 µg BSA, 0.25% glycerol) at room temperature in the dark for 1 h. For supershift experiments, 0.5 µg of anti-Myc antibody (cat. TA150014, Origene) was added 10 min after preincubation of ligands. Samples were then loaded into a 6% native polyacrylamide gel with 0.5% glycerol and run in a 0.5X TBE buffer at 80 V at 4 °C for 45 min. The signal was detected with Typhoon Instrument (GE cat. 00-4277-85 AC, Healthcare) using filters for red light emission detection. 

### 4.9. HuR Protein Expression, Purification, and AlphaScreen with the TNFalpha AU-Rich Element

Recombinant HuR-cMycHis protein preparation and purification and AlphaScreen with a 5′-biotinylated RNA probe (BiTNF, 5′-AUUAUUUAUUAUU UAUUUAUUAUUUA) were carried out as already described [27,67]. Briefly, 1–3 nM of purified recombinant HuR were incubated with 50 nM of a BiTNF probe and AlphaScreen beads (cat. 6760611M, PerkinElmer) at a final concentration of 20 µg/mL. The inhibitory activity of compounds was tested at the indicated concentrations.

### 4.10. Molecular Modeling Studies

A 3D model of LIN28B containing both the cold shock domain (CSD) and the zinc knuckle domain (ZKD) was built from different experimentally solved structures by homology modeling using three different templates. More precisely, the structure CSD of human LIN28B (PDB code: 4A4I, X-ray) [68] and the LIN28-Zinc finger domains bound to AGGAGAU of pre-let-7 miRNA human LIN28A (PDB code: 2LI8, NMR) [69] were selected to cover the segment Val27-Ser176 on LIN28B. The mouse LIN28A structure in the complex with let-7d (PDB code: 3TZR, X-ray) [70] was used to determine the relative orientation between the CSD and ZKD domains. The model was built with the homology modeling routine of the MOE2016 [71] suite using the Amber12 force field [72]. The segment 177–250, suggested to be particularly flexible, was not modeled due to the lack of a reliable template. The selected template guaranteed a confidence superior to 90% according to phyre2 [73].

Molecular docking studies were conducted using Plants1.2 coupled with the ChemPLP scoring function [74]. The binding site was defined as a sphere centered on the pre-let-7 miRNA binding site. Ligand structures were retrieved from PubChem (https://pubchem.ncbi.nlm.nih.gov/, accessed 4 May 2022) using CID 370 (GA), 65064 (EGCG), 169167 (TFMG), and 467299 (EGCDG) and prepared using the MOE2016 wash tools.

### 4.11. EGCG Stability Evaluation by HPLC Analysis

Six concentrations (10, 25, 50, 100, 250, and 500 µM, 1% DMSO each) of EGCG were prepared from the 100 mM stock solution. The calibration curve was constructed by injecting the EGCG dilutions into an Agilent 1200 high-performance liquid chromatography (HPLC) system equipped with an autosampler, binary pump, and diode array detector. A Phenomenex Gemini 5 µm C18 110 Å column (LC Column 250 × 4.6 mm, cat. 00G-4435-E0, Phenomenex, Torrance, CA, USA) was used, and the elution was performed under isocratic conditions with 80:20 water/acetonitrile and 0.01% TFA (pH 4–4.5). The flow rate and the detection were set at 1 mL·min^−1^ and at 280 nm, respectively. The injection volume was 5 µL, and the total run time was set to 15 min at 25.0 °C. Each tested solution was injected within 1 min of its preparation by diluting the DMSO stock solution. The calibration curve obtained by plotting the area of the peaks as a function of the concentration gave an R^2^ of 0.9978. Stability was evaluated using a 50 µM EGCG solution in a cell culture medium and by evaluating the molecule’s stability by HPLC at different time points (0, 15, 30, 45, 60, and 75 min).

### 4.12. NP Preparation

EGCG-NPs, which are composed of a blend of two polymers, poly(epsilon)-caprolactone (PCL) and amine poly(ethylene glycol)-block-poly(lactide-co-glycolide) (PLGA-PEG-NH_2_), were prepared and characterized as previously described [32,75,76]. Briefly, PCL and PLGA-PEG-NH_2_ polymers (mass ratio of 1.5:1), and EGCG (5% *w*/*w*) dissolved in acetonitrile were added dropwise under gentle stirring to a Pluronic F-127 solution (0.1% *w*/*w*), giving a final polymer concentration of 7.0 mg/mL. The resulting suspension was stirred at room temperature to evaporate the organic solvent, then centrifuged and washed to remove the non-encapsulated EGCG. Empty-NP was produced in a similar manner and used for comparison. The dye-loaded NP, i.e., Cou6-NP, was prepared by adding the fluorophore coumarin 6 (cat. 442631, Sigma-Aldrich) 0.05% *w*/*w* instead of EGCG to the polymer solutions.

### 4.13. Detection of Cou6-NPs

NB69 (100.000 cells/well) and CHP134 (150.000 cells/well) were seeded in a six-well plate and treated with different concentrations of Cou6-NPs (NB69: NT, 0.0003 µg/µL, 0.003 µg/µL, and 0.03 µg/µL; CHP134: NT, 0.003 µg/µL) for 48 h. After a wash with PBS, cells were fixed with a paraformaldehyde solution (4% *v*/*v* final, 15 min incubation at room temperature), followed by two washes with PBS. Hoechst 33342 (1 μg/mL, Thermo Fisher Scientific) and HCS CellMask™ Deep Red Stain (Thermo Fisher Scientific, H32721, 1:2000, 20 min at room temperature) were used to identify cell nuclei and cell surfaces, respectively. The fluorescence signal in NB69 cells was detected using the Operetta High Content Imaging System (PerkinElmer) and quantified using the Harmony software 4.1 (PerkinElmer). The CHP134 cells were imaged with a Leica TCS SP8 confocal microscope equipped with a 63×/1.4 oil objective and the proper laser/filter setting. Images were acquired at 400 Hz unidirectional scan speed with 2× zoom and a 130 nm z-step.

### 4.14. Evaluation of EGCG, EGCG-NP, and Empty-NP Treatment on NB Cell Viability and Proliferation

NB cell lines were seeded into 96-well microplates in 100 μL of media. After 24 h, serial dilutions of EGCG, EGCG-NP, and empty-NP were prepared in PBS, and 10 μL of these dilutions were added to the cells and incubated for a specified period of time. The cell viability was evaluated using the CellTiter-Glo^®^ Luminescent Cell Viability Assay (cat. G7570, Promega, Madison, WI, USA) following the manufacturer’s instructions. Depending on the experimental setup, cell viability was expressed either as a percentage of the respective non-treated control or normalized to viability 24 h post-seeding (treatment day 0). Dose–response curves were plotted, and the IC_50_ values were calculated via GraphPad Prism 8.4.2.

### 4.15. Zebrafish Models

Maintenance, breeding, and staging of zebrafish were performed as previously described [77].

Transgenic Tg(fli1:EGFP) zebrafish embryos [33] at 48 h post-fertilization (hpf) were anesthetized using 0.003% tricaine (cat. E10521, Sigma-Aldrich) and carefully positioned on a 10 cm Petri dish containing 3% agarose. SK-N-BE(2) cells, previously treated with 10 μM of EGCG-NPs, or empty-NPs as a comparison, were labeled with a Vybrant^®^ DiL Cell-Labeling Solution (cat. V22885, Thermo Fisher Scientific) following the manufacturer’s guidelines. Fluorescent cells were then resuspended in 1xPBS and implanted using borosilicate glass capillary needles (outer diameter/inner diameter: 1.0/0.75 mm, WPI), a Pneumatic Picopump, and a micromanipulator (WPI). Around 300 cells were injected into the duct of Cuvier of anesthetized embryos. Following implantation, zebrafish embryos were kept at 33 °C. After 4 h post-injection, embryos with fewer than 40 cells were excluded from further analysis. Live photographs of embryos at 2 and 24 h post-implantation (hpi) were captured using a ZeissAxio Observer microscope (Zeiss, Oberkochen, Germany). The absolute fluorescence intensity, expressed as an arbitrary unit, of xenografted NB cells was analyzed and quantified using ImageJ software 1.53a (NHI, Bethesda, MD, USA and University of Wisconsin, Madison, WI, USA).

### 4.16. Statistical Analysis

Results were reported as mean ± SD (standard deviation) or mean ± SEM (standard error of the mean), as indicated in the figure legend. Details of each analysis are in the figure legends. Statistical significance was determined by an unpaired two-tailed *t* test or two-way ANOVA followed by the post hoc Fisher’s LSD test (* *p* < 0.05, ** *p* < 0.01, *** *p* < 0.001, **** *p* < 0.0001; ns *p* ≥ 0.05). For the comparison of heteroscedastic samples, we applied the Welch’s correction to the *t*-test. n represents the number of biological or technical replicates, as indicated in figure legends. All the experiments with representative images (including immunoblotting and immunofluorescence) were repeated at least twice, and representative images are shown.

## Figures and Tables

**Figure 1 ijms-25-04795-f001:**
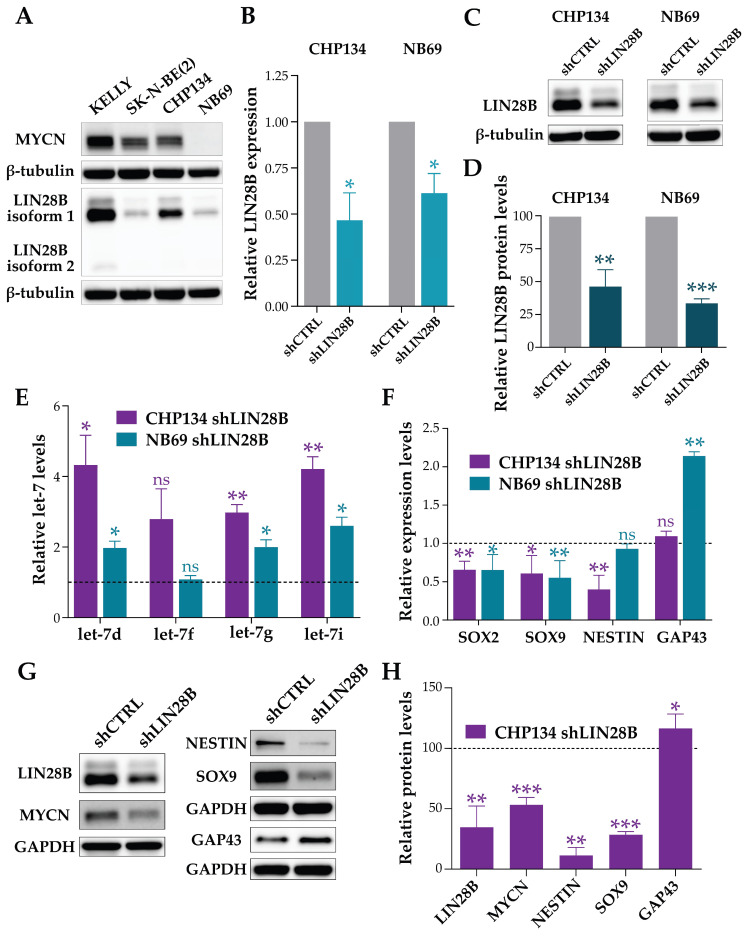
LIN28B downregulation increases let-7 miRNA levels and promotes differentiation in NB cells. (**A**) Representative immunoblot showing MYCN and LIN28B levels in KELLY, SK-N-BE(2), CHP134, and NB69 NB cell lines. β-TUBULIN was used as a loading control. n = 3 biological replicates. (**B**) LIN28B mRNA expression levels analyzed by qPCR in shLIN28B CHP134 and shLIN28B NB69 cell lines. Data were normalized to shCTRL cells. n = 3 biological replicates. Mean ± SD. Unpaired two-tailed Welch’s *t*-test analysis (* *p* < 0.05). (**C**) Representative immunoblots showing LIN28B levels in shLIN28B and shCTRL CHP134 and NB69 cell lines. β-TUBULIN was used as a loading control. (**D**) LIN28B protein levels in shLIN28B CHP134 and shLIN28B NB69 cells. Data were normalized to shCTRL cells. n = 3 biological replicates. Mean ± SD. Unpaired two-tailed Welch’s *t*-test analysis (** *p* < 0.01; *** *p* < 0.001). (**E**) Let-7 miRNA expression levels analyzed by qPCR in shLIN28B CHP134 and shLIN28B NB69 cell lines. Expression levels are shown as fold change relative to shCTRL cells (dashed line). Data were normalized to the internal reference gene U6. n = 2 biological replicates, n = 3 technical replicates each. Mean ± SD. Unpaired two-tailed *t*-test analysis (ns = not significant; * *p* < 0.05; ** *p* < 0.01). (**F**) *SOX2*, *SOX9*, *NESTIN*, and *GAP43* mRNA expression levels analyzed by qPCR in shLIN28B CHP134 and shLIN28B NB69 cell lines. Fold change relative to shCTRL cells is shown (dashed line). *SDHA* was used as an internal reference gene. n = 3 biological replicates, n = 3 technical replicates each. Mean ± SD. Unpaired two-tailed *t*-test analysis (ns = not significant; * *p* < 0.05; ** *p* < 0.01). (**G**) Representative immunoblot showing LIN28B, MYCN, NESTIN, SOX9, and GAP43 levels in shCTRL and shLIN28B CHP134 cells. GAPDH was used as a loading control. n = 3 biological replicates. (**H**) LIN28B, MYCN, NESTIN, SOX9, and GAP43 protein levels in shLIN28B CHP134 cells. Data were normalized to shCTRL cells (dashed line). n = 3 biological replicates. Mean ± SD. Unpaired two-tailed Welch’s *t*-test analysis (* *p* < 0.05; ** *p* < 0.01; *** *p* < 0.001).

**Figure 2 ijms-25-04795-f002:**
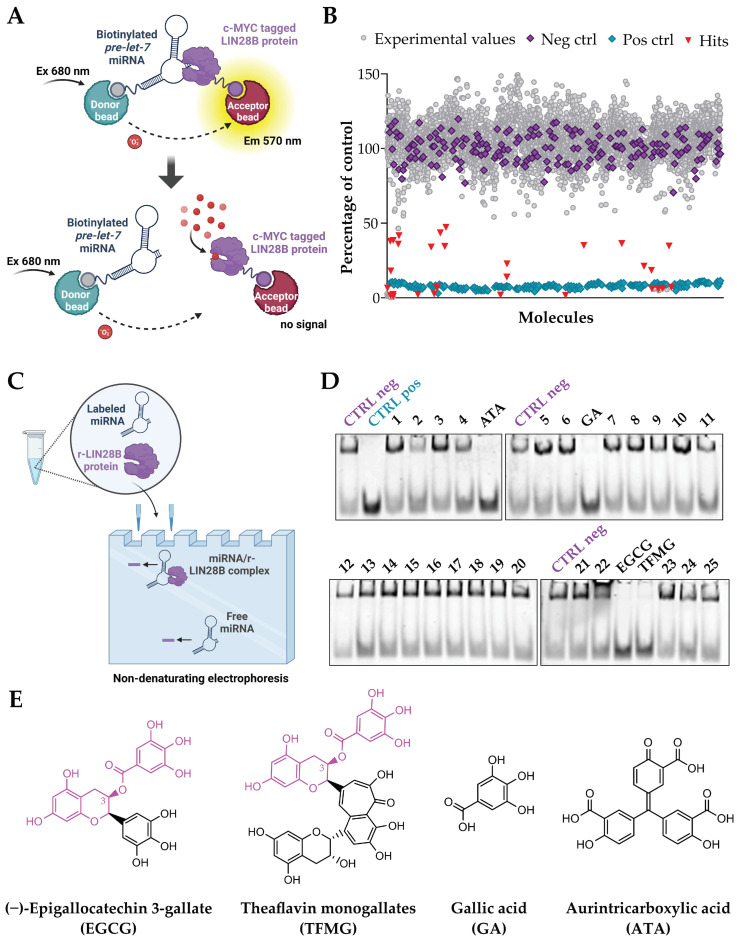
Identification of candidate molecules that interfere with the LIN28B/let-7 interaction: screening and validation. (**A**) Schematic representation of the AlphaScreen technique. (**B**) A dot plot summarizing the screening results expressed as a percentage of the mean of the negative controls. A biotinylated pre-let-7g miRNA was used as a substrate for interaction with the rLIN28B. No drug addition was used as a negative control (highlighted in purple), while a biotinylated pre-let-7g mut miRNA was used instead of the biotinylated pre-let-7g miRNA as a positive control (highlighted in light blue). Compounds that differed by two times the standard deviation from the mean of the negative controls were selected as hits (highlighted in orange). (**C**) Schematic representation of the REMSA. (**D**) Representative REMSA results for the validation of the hits selected by AlphaScreen. The rLIN28B protein plus a Cy3-labelled pre-let-7g miRNA probe was used as a negative control, while the free Cy3-labelled pre-let-7g miRNA probe was used as a positive control. (1) Terbutaline hemisulfate, (2) thioguanine, (3) thioridazine hydrochloride, (4) suramin, (5) diflubenzuron, (6) *N*-hydroxymethylnicotinamide, (7) salicylanilide, (8) dibutyl phthalate, (9) aminosalicylate sodium, (10) amoxicillin, (11) amphotericin B, (12) anthralin, (13) chloramphenicol, (14) chlorcyclizine hydrochloride, (15) dapsone, (16) ethionamide, (17) telenzepine hydrochloride, (18) medroxyprogesterone acetate, (19) piperazine, (20) procaine hydrochloride, (21) acedapsone, (22) doxorubicin, (23) dehydro (11,12)ursolic acid lactone, (24) coralyne chloride, (25) 2′,5′-dihydroxy-4-methoxychalcone. (**E**) Molecular structures of EGCG, TFMG, GA, and ATA. TFMG was present in the screened library as mixed isomers from black tea. Theaflavin 3-gallate isomer is shown here. The degree of structural similarity between EGCG and TFMG is highlighted in pink.

**Figure 3 ijms-25-04795-f003:**
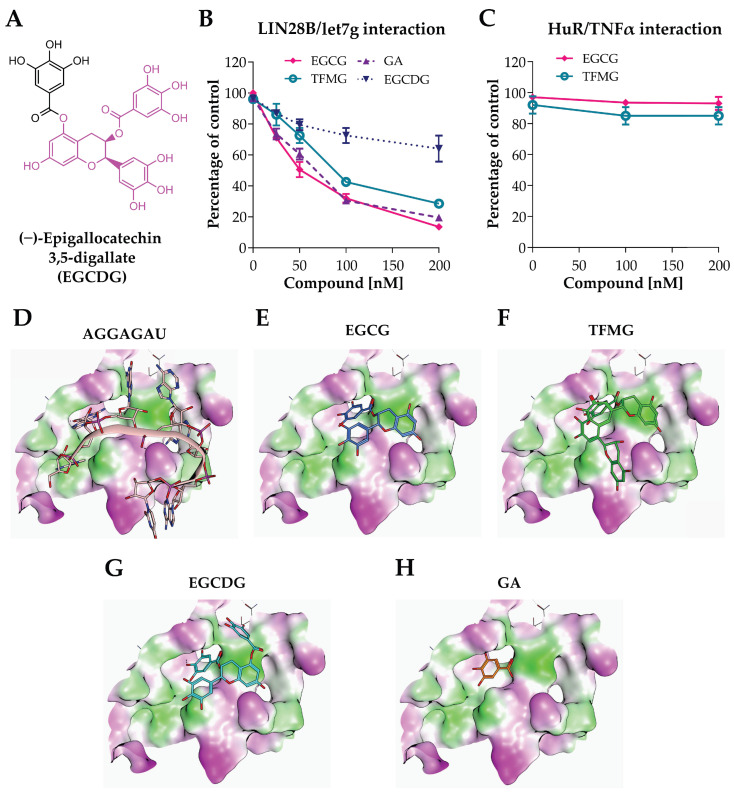
EGCG disrupts the LIN28B/let-7 interaction by binding to LIN28B. (**A**) Molecular structure of (−)-epigallocatechin 3,5-digallate (EGCDG). The degree of structural similarity with EGCG and TFMG is highlighted in pink. (**B**) Dose-dependent titration experiments performed using the Alpha assay showing the effect of increasing concentrations of EGCG, TFMG, GA, and EGCDG on the interaction between rLIN28B and biotinylated pre-let-7g miRNA. (**C**) Dose-dependent titration experiments performed using the Alpha assay showing the effect of increasing concentrations of EGCG and TFMG on the interaction between HuR and the TNFα AU-rich element. (**D**–**H**) Molecular modeling studies: the binding conformation of the pre-let-7 miRNA (AGGAGAU) (**D**) and the obtained poses by molecular docking for EGCG (**E**), TFMG (theaflavin 3-gallate) (**F**), EGCDG (**G**), and GA (**H**).

**Figure 4 ijms-25-04795-f004:**
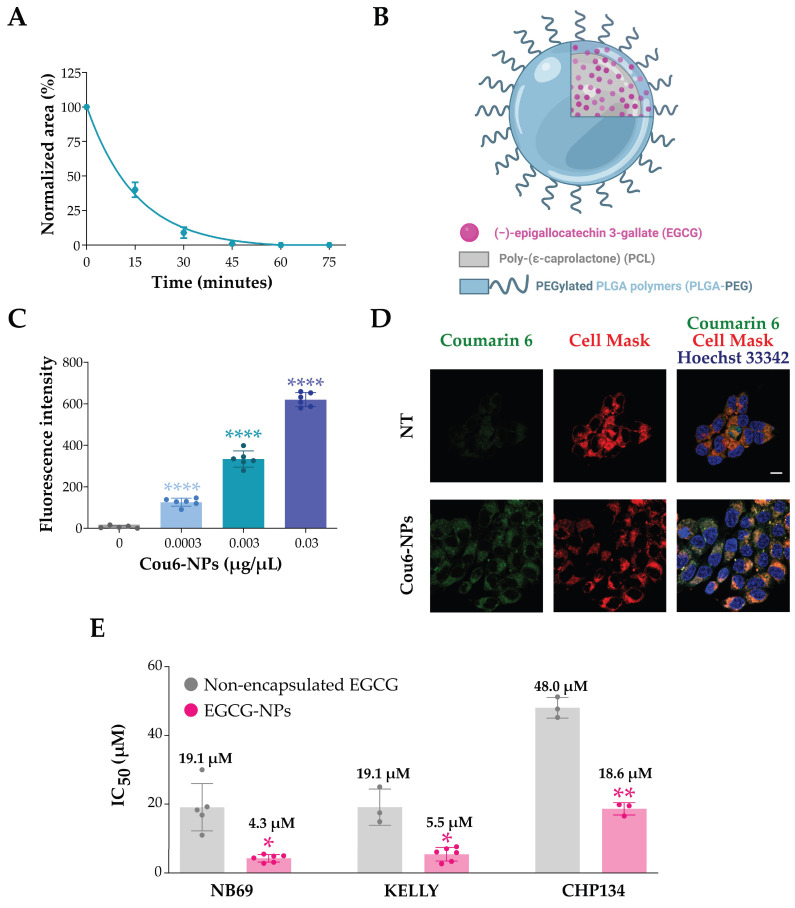
Evaluation of EGCG stability by HPLC, schematic representation of the structure of the EGCG-NPs, and assessment of their penetration and effect in NB cells. (**A**) Percentage of EGCG amount over time (0, 15, 30, 45, 60, and 75 min) under cell culture conditions. The area of the EGCG peak was normalized to t = 0. n = 3 replicates, mean ± SD. (**B**) Schematic representation of the EGCG-NPs. Adapted from [32]. (**C**) Immunofluorescence analysis of NB69 cells treated with different amounts of Cou6-containing NPs. Images acquired using the Operetta-High Content Imaging System were analyzed using the Harmony software 4.1, and the average fluorescence intensity of the Cou6 fluorescent dye (green) was quantified. n = 6 technical replicates. Two-way ANOVA followed by Fisher’s LSD test (**** *p* < 0.0001). (**D**) Representative confocal images of CHP134 cells treated with 0.003 µg/µL of Cou6-NPs (green). Nuclei were stained with Hoechst 33342 (blue), and cytoplasm was stained with the CellMask™ Deep Red Stain (red). Scale bar = 10 µm. (**E**) IC_50_ values for NB69, KELLY, and CHP134 after treatment with non-encapsulated EGCG or EGCG-NPs. n = at least 3 biological replicates, n = 3 technical replicates each. Mean ± SD. Unpaired two-tailed Welch’s *t*-test (* *p* < 0.05, ** *p* < 0.01).

**Figure 5 ijms-25-04795-f005:**
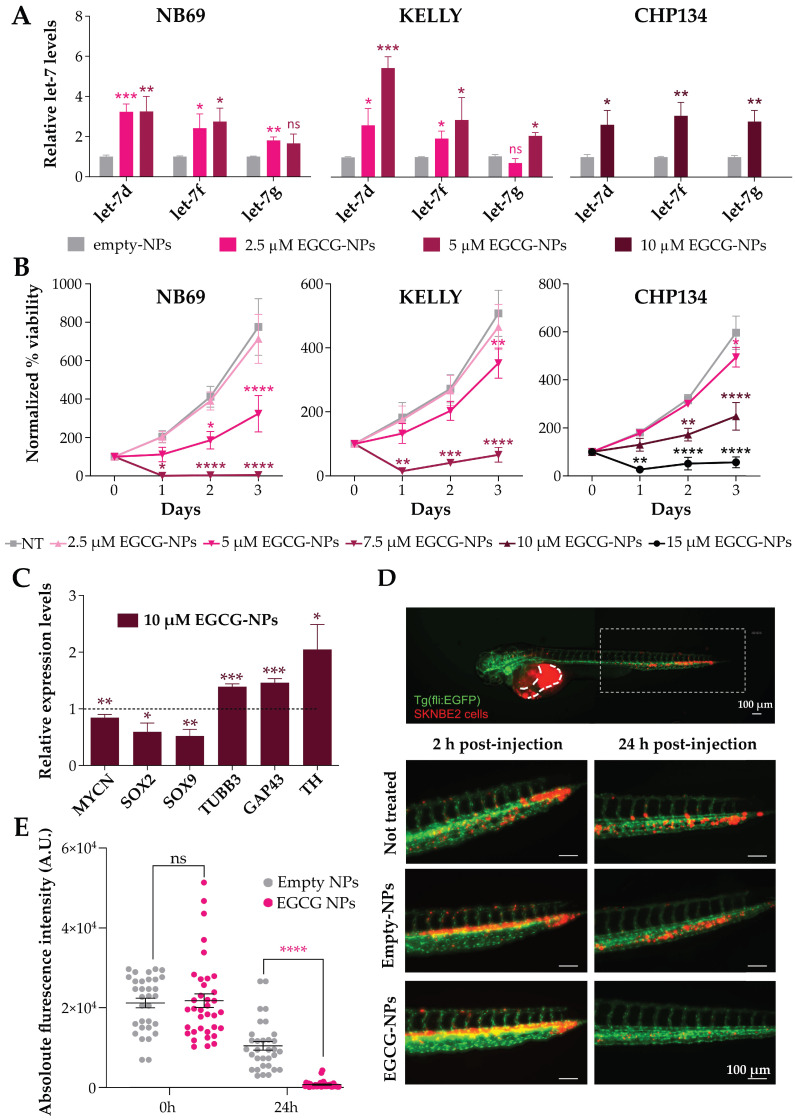
EGCG-NP treatment increases let-7 miRNA levels, decreases proliferation, promotes differentiation in NB cells, and reduces their engraftment ability in zebrafish. (**A**) qPCR analysis of let-7d, let-7f, and let-7g miRNAs in NB69, KELLY, and CHP134 cell lines treated for 48 h with empty-NPs or EGCG-NPs. The expression level is shown as fold change relative to non-treated cells, and the data were normalized to the internal reference gene U6. n = 3 biological replicates, n = 3 technical replicates each. Mean ± SD. Unpaired two-tailed *t*-test analysis (* *p* < 0.05; ** *p* < 0.01; *** *p* < 0.001, ns *p* ≥ 0.05). (**B**) Proliferation curves of NB69, KELLY, and CHP134 cells treated with different doses of EGCG-NPs around or below the IC_50_ values. Cell viability was measured using the CellTiter-Glo^®^ Luminescent Cell Viability Assay and normalized to the day of the treatment (day 0). NT = non-treated cells. n = 3 biological replicates, n = 3 technical replicates each. Mean ± SEM. Two-way ANOVA followed by Fisher’s LSD test (* *p* < 0.05, ** *p* < 0.01, *** *p* < 0.001, **** *p* < 0.0001). (**C**) *MYCN*, *SOX2*, *SOX9*, *TUBB3*, *GAP43*, and *TH* mRNA expression levels analyzed by qPCR after 96 h of EGCG-NP treatment in CHP134 cells. Expression level is shown as fold change relative to empty-NP treatment (dotted line). *SDHA* was used as a reference gene. n = 3 biological replicates, n = 3 technical replicates each. Mean ± SD. Unpaired two-tailed *t*-test analysis (ns = not significant; * *p* < 0.05; ** *p* < 0.01, *** *p* < 0.001). (**D**) Upper panel shows a transgenic zebrafish embryo with NB cells (red signal) at the injection site in the duct of Cuvier (dashed lines) and in the caudal region (white dashed square). Representative fluorescence microscopy images of the caudal region of Tg(fli1:EGFP) zebrafish embryos analyzed 2 h and 24 h after injection with non-treated or pretreated SK-N-BE(2) cells (empty-NPs and EGCG-NPs) and labeled with the Vybrant^®^ DiI (red signal). Scale bar = 100 µm. (**E**) Absolute fluorescence intensity of SK-N-BE(2) cells pretreated with either empty-NPs or EGCG-NPs measured at time 0 or 24 h after injection. Each dot represents the value from a single embryo. A.U. = arbitrary units. Mean ± SEM. n ≥ 32 zebrafish embryos analyzed per condition. Unpaired two-tailed *t*-test analysis (**** *p* < 0.0001).

## Data Availability

Data are available upon request to the corresponding authors.

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
