# Peer review of "EGCG Disrupts the LIN28B/Let-7 Interaction and Reduces Neuroblastoma Aggressiveness"

_ijms, 2024, doi:10.3390/ijms25094795_

Round 1

Reviewer 1 Report

Comments and Suggestions for Authors

In this manuscript, Simona Cocchi and colleagues showed that Epigallocatechin-3-gallate (EGCG) disrupts the LIN28B/let-7 interaction and reduces neuroblastoma aggressiveness. This study is novel, and the manuscript is well-written; I have some minor concerns. 

References are missing in some of the statements; for example-  "Neuroblastoma (NB) is the most frequently diagnosed extracranial solid tumor in children. The majority of cases occur in patients under the age of 5, with an average age of diagnosis of 2 years."

"NBs arise from the developing peripheral sympathetic nervous system in the adrenal medulla or along the sympathetic chain." 

"The paralog LIN28B is recognized as an oncogene in NB and plays an important role in NB tumorigenesis."

Figure 1C, putting the blot image first and then their quantifications will be good.

Author Response

Thank you for your valuable suggestions.

Comment 1: References are missing in some of the statements; for example-  "Neuroblastoma (NB) is the most frequently diagnosed extracranial solid tumor in children. The majority of cases occur in patients under the age of 5, with an average age of diagnosis of 2 years."

"NBs arise from the developing peripheral sympathetic nervous system in the adrenal medulla or along the sympathetic chain." 

"The paralog LIN28B is recognized as an oncogene in NB and plays an important role in NB tumorigenesis."

Response: Thank you for this suggestion. We have added the appropriate references at the end of each sentence.

Comment 2: Figure 1C, putting the blot image first and then their quantifications will be good.

Response: Thank you for this suggestion. We have divided Figure 1C and have shown the blot image first (new Figure 1C) and then their quantification (new Figure 1D). We have then changed the number of the other Figures accordingly (Figure 1E-H).

The file attached contains the response to all reviewers comments.

Reviewer 2 Report

Comments and Suggestions for Authors

The authors present a clear case of how EGCG in a nanoparticle formulation can disrupt the Lin28B-Let7 interaction which has a detrimental effect on neuroblastoma cells. The manuscript is clear and well written and present a potentially interesting new treatment avenue for neuroblastoma. 

One, relatively minor, issue I have is that the authors suggest multiple times that EGCG affects differentiation, however, they base this on the overexpression of just one gene (at least in the first figure). They should show more established genes related to differentiation that are upregulated to really make this claim. Moreover, they mention stemness genes that are downregulated, but this is a vague term in my view, with no real solid basis in neuroblastoma tuor biology, so I would prefer a more neutral term like tumor markers. 

Finally, I have several comments/suggestions regarding the figures:

- Fig. 1B,C there are no error bars in the controls. I assume they are also based on triplicates, so they should show error bars

- Fig 2B, it would be interesting to color the compounds selected fro further characterization

-Fig. 4E, data would be more clearly presented in a boxplot instead of a barplot. Might be relevant for more figures. 

And some small comments regarding the text:

- Line 238 ("demonstrating that both compounds are selective antagonists of the 238 LIN28B/RNA interaction") is too strongly worded after one experiment showing that a different interaction is not affected. 

- In the discussion I miss some more discussion on whether and how this formulation can be used in patients, and which neuroblastoma patients are expected to benefit (all? subgroups?). 

Comments on the Quality of English Language

No. 

Author Response

Thank you for your valuable suggestions.

Comment 1: One, relatively minor, issue I have is that the authors suggest multiple times that EGCG affects differentiation, however, they base this on the overexpression of just one gene (at least in the first figure). They should show more established genes related to differentiation that are upregulated to really make this claim. Moreover, they mention stemness genes that are downregulated, but this is a vague term in my view, with no real solid basis in neuroblastoma tuor biology, so I would prefer a more neutral term like tumor markers. 

Response: Thank you for this comment, which we agree with in part. 

Regarding Figure 1, we have used the more general concept of reducing tumor aggressiveness instead of promoting differentiation. 

We have also replaced the term "stemness markers" with "tumour markers" throughout the manuscript, as you suggest. 

Regarding EGCG treatment and differentiation promotion (Figure 5C), we have left in the Results section only the description of what we observed (a decrease in the tumor markers MYCN, SOX2 and SOX9 and an increase in the differentiation markers TH, TUBB3, GAP43) without further speculation. 

However, we have left the suggestion that EGCG may promote NB differentiation in the Discussion section. Indeed, our data are supported by literature data on let-7 miRNAs, which have been described as fundamental tumor suppressors and essential regulators of terminal differentiation (Lee H, et al. Biogenesis and regulation of the let-7 miRNAs and their functional implications. Protein Cell. 2016 Feb;7(2):100-13). Therefore, the increase in let-7 miRNAs observed with EGCG treatment is likely to induce differentiation, which is also supported by our data on the decrease in tumor markers and increase in differentiation markers.

We have added in the Discussion section the following sentence (lines 484-486): “Given the role of let-7 miRNAs, which have been described as fundamental tumour suppressors and essential regulators of terminal differentiation [60], and the effects we observed upon EGCG treatments, EGCG treatment may promote NB cell differentiation”.

Comment 2: Fig. 1B,C there are no error bars in the controls. I assume they are also based on triplicates, so they should show error bars

Response: Thank you for your comment. In Figure 1B and 1C (now Figure 1D in this new version according to reviewer’s 1 suggestion) the experimental values for the shCTRL cells have been normalized to 1 and 100, respectively. The data represent the mean ± SD of three biological replicates as reported in the figure legend. Since in each biological replicate the shCTRL value has been normalized to 1 or 100, the SD for the three biological replicates is equal to 0, therefore there are no error bars reported.

To show a standard deviation to the aforementioned shCRTL bars, we would have to show the raw data, but the data from types of experiments are commonly shown as a percentage of control and we prefer to use this type of representation.

Another option would be to remove the shCTRL bar and show just the shLIN28B bar, such as in Figures 1E, 1F, and 1H. We can do this if you think this representation would be clearer.

Comment 3: Fig 2B, it would be interesting to color the compounds selected for further characterization.

Response: Thank you for the suggestion. The change has been implemented.

Comment 4: Fig. 4E, data would be more clearly presented in a boxplot instead of a barplot. Might be relevant for more figures

Response: Thank you for the suggestion. The change has been implemented for Figure 4E, as well as for Figures 4C and 5E.

Comment 5: Line 238 ("demonstrating that both compounds are selective antagonists of the 238 LIN28B/RNA interaction") is too strongly worded after one experiment showing that a different interaction is not affected

Response: Thank you for pointing this out, we agree with the comment. We have therefore softened the tone of the sentence by removing "demonstrating that both compounds are selective antagonists of the 238 LIN28B/RNA interaction".

Comment 6: In the discussion, I miss some more discussion on whether and how this formulation can be used in patients, and which neuroblastoma patients are expected to benefit (all? subgroups?). 

Response: Thank you for the comment. We have integrated the Discussion section with:

  • information on the advancement of the use of nanomedicine in NB. Lines 451-461: To date, most studies on the application of nanomedicine strategy to NB therapeutics have been conducted mainly in the preclinical setting using cellular and animal experiments, which together have provided some positive evidence [55,56]. Despite these encouraging results, the therapeutic potential of nanomedicine in NB has not yet been systematically explored and only albumin-bound paclitaxel nanoparticles (i.e. Abraxane) have reached phase I/II clinical trials for refractory NB and other pediatric solid tumors (NCT01962103) [57,58].In this scenario, among the different approaches pursued in the field of nanoformulation, we chose to use polymeric nanoparticles as a model of nanosystems for ECGC delivery.”
  • information about the limits of EGCG use in the clinic. Lines 489-498: “The poor pharmacokinetic profile of EGCG, which requires the use of a nanocarrier-based formulation, represents the main limitation of the present study and potentially hinders a streamlined clinical development. Future studies aimed at elucidating the structure of EGCG bound to LIN28B could guide the design of new molecules with greater potency and better pharmacokinetic properties, with the ultimate aim of selecting a molecule more suitable for further preclinical and clinical development. In this context, EGCG should be considered as a chemical probe to prove that inhibition of the LIN28B/let-7 axis is a novel and promising therapeutic option for NB, especially with regard to the development of new agents for differentiation therapies.”
  • information on which patients could benefits from a differentiation therapy approach in NB. Lines 505-515: “In NB, 13-cis-retinoic acid, a pro-differentiating agent, is currently used in clinical practice as part of the treatment of patients with high-risk NB in the post-consolidation phase of the therapeutic schedule [62–64]. Indeed, the rate of tumor relapse is directly dependent on the efficacy of post consolidation. Unfortunately, many patients are refractory to retinoic acid-induced differentiation and further research efforts, including synergistic combination therapy, are needed [65,66]. In this context, EGCG and, more generally, inhibitors of the LIN28B/let-7 miRNA circuit may represent good candidates to be used as experimental drugs in the post consolidation phase for high-risk patients, both alone and as part of synergistic pro-differentiating multi-drug treatments.”

The file attached contains the response to all reviewers comments.

Reviewer 3 Report

Comments and Suggestions for Authors

Submitted manuscript presents results of study aiming to target LIN28B / let-7 interaction in neuroblastoma (NB) cell lines and evaluate significance of this interaction for NB cells viability/aggressiveness. The authors screened almost 5000 of molecules for their ability to disrupt LIN28B-let-7 interaction and then delivered the selected particles to the NB cells using nanoparticle carriers. The study provides novel and interesting data on novel treatment strategies in NB. The manuscript is well written, the data are clearly presented. However, there are some improvements that shall be considered in revised version of the manuscript before it could be considered for publication. 

1. The discussion is very concise. The authors shall consider better analysis and discussion of study limitations and the perspectives and aims for the future studies. E.G. Delivery of LIN28B/let-7 inhibitor EGCG to the NB cell lines in vitro included nanocarriers, partially because of its short half-life in culture media and poor solubility. LIN28B protein can affect cancer cells, including NB cells, in let-7 independent manner but and it shall be also considered as potential limitation. Long terms effects, drug resistance and other relevant limitations shall be better addressed in the discussion chapter.

2. The reasons for selection of specific NB cell lines for some experiments are unclear and more explanation in the Results and/or discussion chapters could be helpful. 

3. Name/s of cell line/s used for experiments shall be added to each graph. For example Figure 5C presents results derived from CHP134 cells while Figure 5E – SK-N-BE(2) cells. Names of cell lines are clearly indicated in Figures 5 A and B, barely visible in 5 D and not given in Figure 5 C and 5 E. 

4. Manufacturers/material provides shall be referenced in the material and methods chapter with full details, including city, state/province if applicable and country. Many manufacturers are not given or they affiliation misses these details. 

5. Line 486-487 – concentration of streptomycin (10 mg streptomycin/mL) is missing.

6. Line 496 – 45 or 0.45 micro meters?

7. Lines 513-515 – quantification method shall be referenced here. 

8. Line 520 -  amount of proteins in micro grams per well shall be provided

9. The working dilutions, time and temperature of incubations for all primary and secondary antibodies used in the study shall be provided. 

10. Statistical methods used in the study shall be summarized in a subsection of material and methods. 

Author Response

Thank you for your valuable suggestions.

Comment 1: The discussion is very concise. The authors shall consider better analysis and discussion of study limitations and the perspectives and aims for the future studies. E.G. Delivery of LIN28B/let-7 inhibitor EGCG to the NB cell lines in vitro included nanocarriers, partially because of its short half-life in culture media and poor solubility. LIN28B protein can affect cancer cells, including NB cells, in let-7 independent manner but and it shall be also considered as potential limitation. Long terms effects, drug resistance and other relevant limitations shall be better addressed in the discussion chapter.

Response: Thank you for the comment. We have integrated the Discussion section with:

  • a comment on LIN28B pro-tumorigenic activity independent from let-7 miRNAs. Lines 475-480: “Of note, LIN28B has been shown to have pro-tumorigenic activity independent of its interaction with let-7 miRNAs by binding to specific mRNAs and acting as a post-transcriptional regulator [59]. Further experiments to investigate whether EGCG can also affect LIN28B pro-tumorigenic activity in a let-7 independent manner are definitely needed and may strengthen the relevance of EGCG in NB and potentially other tumor types.”
  • a comment on the limitations of the study and future aims. Lines 489-498: The poor pharmacokinetic profile of EGCG, which requires the use of a nanocarrier-based formulation, represents the main limitation of the present study and potentially hinders a streamlined clinical development. Future studies aimed at elucidating the structure of EGCG bound to LIN28B could guide the design of new molecules with greater potency and better pharmacokinetic properties, with the ultimate aim of selecting a molecule more suitable for further preclinical and clinical development. In this context, EGCG should be considered as a chemical probe to prove that inhibition of the LIN28B/let-7 axis is a novel and promising therapeutic option for NB, especially with regard to the development of new agents for differentiation therapies.

Comment 2: The reasons for selection of specific NB cell lines for some experiments are unclear and more explanation in the Results and/or discussion chapters could be helpful. 

Response: Thank you for your comment. We have included an explanation on the use of the specific cell lines in the text.

Specifically, we started our project by analyzing four NB cell lines (Kelly, SK-N-BE(2), CHP134 and NB69) for LIN28B and MYCN expression levels and selected two of them (CHP134 and NB69, which express different levels of LIN28B and have different MYCN amplification status) to downregulate LIN28B expression. We then continued with the two selected cell lines (CHP134 and NB69) and decided to include the KELLY cell line as it harbors extremely high amplification levels of the MYCN locus [24] and expresses high levels of LIN28B (Figure 1A). Lines 302-305: "Specifically, we used the CHP134 and NB69 cell lines that we used for LIN28B downregulation and the KELLY cell line which harbors extremely high amplification levels of the MYCN locus [24] and expresses high levels of LIN28B (Figure 1A)”.

We continued to use these 3 NB cell lines to evaluate the effects of EGCG-NP treatment on let-7 miRNA level variation and cell growth. Next, we used CHP134 to evaluate the effects of EGCG-NP treatment on tumour and differentiation marker variation, as this cell line has been reported to be prone to differentiation upon certain stimuli, such as 13-cis-retinoic acid treatment. Line 348: "Furthermore, EGCG-NP treatment in CHP134 cells, which have been reported to be prone to differentiate upon specific stimuli such as 13-cis-retinoic acid treatment [33], led to a significant ...."

Finally, to investigate whether EGCG-NP pre-treatment would affect the tumourigenic potential of NB cells in vivo in a zebrafish model, we used the SK-N-BE(2) cell line, since the CHP134 cell line does not possess a strong engraftment capacity (data not shown) and no data are available for KELLY and NB69 cells, we used the SK-N-BE(2) cell line.

Lines 361-362: “Since CHP134 cells do not possess a strong engrafting ability (data not shown) and no data are available for KELLY and NB69 cells, we used the SK-N-BE(2) cell line, whose engraftment capacity has already been reported [34].

Comment 3: Name/s of cell line/s used for experiments shall be added to each graph. For example Figure 5C presents results derived from CHP134 cells while Figure 5E – SK-N-BE(2) cells. Names of cell lines are clearly indicated in Figures 5 A and B, barely visible in 5 D and not given in Figure 5 C and 5 E. 

Response: Thank you for this comment. The information on the cell lines used for the experiment is clearly stated in Figures 5A and B because 3 different cell lines were used in these experiments and therefore we had to associate each graph with the specific cell line used. For Figures 5C-E, since it is easy to understand from the main text and/or the figure legend which cell line was used for the experiment, we believe that adding this information to the graph would make the graph more confusing, so we prefer not to add it.

Comment 4: Manufacturers/material provides shall be referenced in the material and methods chapter with full details, including city, state/province if applicable and country. Many manufacturers are not given or they affiliation misses these details. 

Response: Thank you for your comment. We have added the catalogue number and manufacturer's name for each reagent used and the manufacturer/material supplier details when we first mentioned them.

Comment 5: Line 486-487 – concentration of streptomycin (10 mg streptomycin/mL) is missing.

Response: Thank you for your comment. We have added the concentration of the stock reagent. 

Comment 6: Line 496 – 45 or 0.45 micrometres?

Response: Thank you for your comment. We have corrected the value to 0.45 micrometres.

Comment 7: Lines 513-515 – quantification method shall be referenced here. 

Response: Thank you for the suggestion. We briefly expanded this section of Materials and Methods to underline that we used the ΔΔCt method for quantification and we specified the reference genes and internal calibrators that we used.

Comment 8: Line 520 -  amount of proteins in micrograms per well shall be provided

Response: Thank you for your comment. We have added this information.

Comment 9: The working dilutions, time and temperature of incubations for all primary and secondary antibodies used in the study shall be provided. 

Response: Thank you for your comment. We have added this information.

Comment 10: Statistical methods used in the study shall be summarized in a subsection of material and methods. 

Response: Thank you for this suggestion. We have added a subsection called “4.16 Statistical analysis” to the Material and Methods section.

The file attached contains the response to all reviewers comments.
